# Exploring the Practicality of Generative Retrieval on Dynamic Corpora

Soyoung Yoon[*]
KAIST AI
Republic of Korea
soyoungyoon@kaist.ac.kr

Chaeeun Kim[*†]
KAIST AI
Republic of Korea
chaechaek1214@gmail.com

Hyunji Lee
KAIST AI
Republic of Korea
hyunji.amy.lee@kaist.ac.kr

Joel Jang
KAIST AI
Republic of Korea
joeljang@kaist.ac.kr

Sohee Yang
KAIST AI
Republic of Korea
sohee.yang@kaist.ac.kr

Minjoon Seo
KAIST AI
Republic of Korea
minjoon@kaist.ac.kr

## ABSTRACT

Benchmarking the performance of information retrieval (IR) is mostly conducted with a fixed set of documents (static corpora). However, in realistic scenarios, this is rarely the case and the documents to be retrieved are constantly updated and added. In this paper, we focus on Generative Retrievals (GR), which apply autoregressive language models to IR problems, and explore their adaptability and robustness in dynamic scenarios. We also conduct an extensive evaluation of computational and memory efficiency, crucial factors for real-world deployment of IR systems handling vast and ever-changing document collections. Our results on the StreamingQA benchmark demonstrate that GR is more adaptable to evolving knowledge (+ $4 - 11\%$), robust in learning knowledge with temporal information, and efficient in terms of inference FLOPs (×2), indexing time (×6), and storage footprint (×4) compared to Dual Encoders (DE), which are commonly used in retrieval systems. Our paper highlights the potential of GR for future use in practical IR systems within dynamic environments.

## KEYWORDS

Generative Information Retrieval, Corpora Adaptation, Dynamic Corpora, Continual Pretraining

## 1 INTRODUCTION

Transformer-based information retrieval (IR) models play a vital role in advancing the field of semantic document search for information-seeking queries. Notably, *Generative Retrieval* (GR) [3, 7, 23, 24, 27, 38, 42–44, 50] has recently gained a significant amount of recognition from the research community for its simplicity and high performance. However, *Dual Encoder* (DE) [11, 13, 15, 19, 35, 40] continues to hold sway in practical IR systems. This contrast underscores the need for an investigation into their practical applicability. There is a lack of comprehensive comparison between GR and DE in real-world scenarios where knowledge is continually evolving and efficiency is crucial.

To this end, we establish Dynamic Information Retrieval (DynamicIR), a setup designed to simulate realistic scenarios for corpus updates in IR. This DynamicIR setup includes two distinct update strategies, (1) indexing-based updates: updating only the index without any further pretraining or finetuning and (2) training-based updates: continually pretraining the parameters on new corpora in addition to updating the index (See Figure 1). Within this experimental setup, we evaluate the adaptability of recent state-of-the-art retrieval models: Seal [3] and Minder [27] for GR, and Spider [40], Contriever [15], and DPR [19] for DE. Furthermore, we perform extensive comparison for the *efficiency* of each method, considering factors such as floating-point operations (FLOPs) [18] required for the inference, indexing time, inference latency, and storage footprint.

The findings of our study underscore the strength of GR compared to DE across three key aspects: adaptability, robustness, and efficiency. (1) *GR exhibits superior adaptability to evolving corpora* (Section 5.1). GR outperforms DE, showcasing *4 – 11% greater adaptability* in both indexing-based and training-based updates. Notably, GR not only acquires new knowledge more effectively but also show no sign of forgetting; rather, training with new corpora appears to enhance its existing knowledge. (2) *GR is more robust without inducing undesired bias from data characteristics* (Section 5.2). DE reveals a bias towards lexical overlap of timestamps inserted into queries and documents, showing significant degradation (52.23% → 17.40%) when the timestamps are removed. Whereas, GR shows robust retrieval performance over temporal data. (3) *GR requires lower indexing costs, inference flops, and storage footprint* (Section 6).

[*]Both authors contributed equally to this research.
[†]Work done during internship at KAIST AI

[0]Soyoung Yoon initiated the exploration of corpora adaptation for unseen data, trained the DE baselines, Spider, and Contriever, and proposed using StreamingQA and generated pseudo-queries for $R'_{new}$. Chaeeun Kim proposed DynamicIR with two update scenarios, trained GenIR models, SEAL, MINDER, and LTRGR, exploring their adaptability, and analyzed key parameters for adapting new corpora, applying these during continual pretraining. She also experimented on temporal bias, introduced practical aspects in Section 6, and wrote most of this paper. Hyunji Lee, Joel Jang, and Sohee Yang analyzed experimental results, dataset, and model behavior, such as acquisition of time information, temporal or length biases, and calculating FLOPs. Minjoon Seo advised on the paper's direction, suggesting corpus updates/addition scenarios and the use of efficient pretraining methods like LoRA, and analyzing model specificity and experimental results.

Soyoung Yoon,    Chaeeun Kim[*], Hyunji Lee, Joel Jang, Sohee Yang, and Minjoon Seo

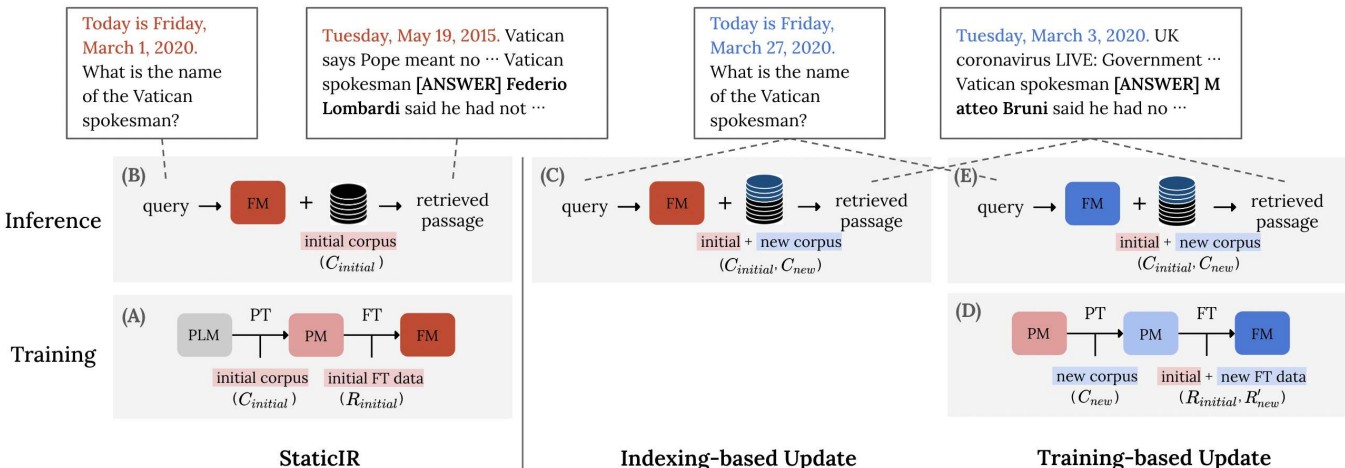

**Figure 1: Structure of DynamicIR.** This figure shows the training and inference processes for three setups in DynamicIR. We differentiate each model by color. First, in StaticIR, (A) retrieval models are pretrained on $C_{initial}$ and finetuned on the query-document pairs $R_{initial}$. (B) During inference, they perform retrieval only with the indexed $C_{initial}$. Second, in Indexing-based Update, (C) we use the same retriever developed from StaticIR and conduct an inference with the indexed $C_{initial}$ and $C_{new}$. Lastly, in Training-based Update, (D) we take the pretrained model on $C_{initial}$ in StaticIR and continually pretrain it on $C_{new}$. Subsequently, it is finetuned on the combination of $R_{initial}$ and $R'_{new}$. (E) Using the updated retrieval model, we conduct an inference with the indexed $C_{initial}$ and $C_{new}$.

For inference flops, GR has $O(1)$ complexity with respect to the corpus size, requiring *2 times* less computation per query compared to DE which has $O(N)$ complexity, where $N$ represents the corpus size. Regarding indexing, DE necessitates re-indexing each time whenever the model is updated. To make matter worse, the indexing time itself is *6 times* longer than GR. In terms of storage footprint, GR requires *4 times* less storage by effectively compressing the knowledge in its internal parameters.

## 2 RELATED WORK

*Temporal Information Retrieval.* Temporal information retrieval (IR) [17] has long been a subject of interest in the field of IR. While recent advancements have focused on the temporal updating of *language models* [10], the attention on temporal *IR* has diminished somewhat with the rise of transformer-based models such as BERT [9]. These models are renowned for their robust contextualized embeddings, which likely shifts the focus away from knowledge updates. However, temporal considerations in IR remain crucial, especially with the widespread use of Retrieval-Augmented Generation (RAG) in many chat models. It is also worthwhile to examine whether emerging IR approaches are effective in adapting to evolving knowledge. Unlike previous works on document updates [4, 34], which pose disjoint questions from existing ones when retrieving updated documents and require parameter updates, our approach conducts experiments on the retrieval of distinct documents for the same query across varying timestamps. Similar to the text box in Figure 1, users often want to search for information based on a specific time period (e.g., laws, national curriculum, etc.). We also consider two realistic update scenarios, both with

and without parameter updates and use a large-scale corpus of 50 million passages.

*Generative Retrieval (GR).* GR initially emerged with the work of GENRE [7], in which an encoder-decoder model retrieves a document by generating the title of the document from a given query. [43] introduces DSI that produces a document ID as the output sequence. NCI [44] and DSI-QG [51] apply query generation, significantly improving DSI's performance. Recent methods that uses document ID, such as RIPOR [48] and PAG [49], also demonstrate superior performance. Instead of mapping to IDs for document identifiers, other works explore generating content directly from documents as identifiers. For instance, SEAL utilizes spans [3], and MINDER [27] and LTRGR [26] leverage a combination of titles, pseudo-queries, and spans. Other works focus on the broader application of GR, such as multi-hop reasoning [24], contextualization of token embeddings of retriever [23], auto-encoder approach for better generalization [42], and giving ranking signals [26]. Our work employs GR that utilizes document content as identifiers for temporal information retrieval, as it can directly access the contents and update the pieces of knowledge.

*Dual Encoder (DE).* DE [20, 25] refers to a set of model architectures where we project the query and document individually into a fixed sized embedding. Through contrastive learning, the projected embeddings of positive documents are learned to be close to the query and negative documents to be far away. Some works try to train the model in an unsupervised fashion with contrastive learning [15, 25, 41]. Although external modules such as FAISS [16], and ANCE [47] can help the efficiency of those models in inference

| Type | Split | Count |
|---|---|---|
| Query-Doc pairs | $R_{initial}$ (2007 − 2019) | 99,402 |
| | $R'_{new}$ (2020) | 90,000 |
| Evaluation | $Q_{initial}$ (2007 − 2019) | 2,000 |
| | $Q_{new}$ (2020) | 3,000 |
| | $Q_{total}$ (2007 − 2020) | 5,000 |
| Corpus | $C_{initial}$ (2007 − 2019) | 43,832,416 |
| | $C_{new}$ (2020) | 6,136,419 |
| | $C_{total}$ (2007 − 2020) | 49,968,835 |
| # Tokens | Initial (2007 − 2019) | 7.33B |
| | New (2020) | 1.04B |
| | Total (2007 − 2020) | 8.37B |
| # Tokens per passage | Initial (2007 − 2019) | 169.7 |
| | New (2020) | 167.1 |
| | Total (2007 − 2020) | 167.5 |

**Table 1: Statistics of the StreamingQA dataset modified for our setup. # Tokens is the total number of words separated by space in each passage.**

time, these types of models still fall into the limitation that model-dependent embedding dumps need to be made in an asynchronous fashion.

## 3 DYNAMIC INFORMATION RETRIEVAL

### 3.1 DynamicIR Task Setup

Adapting the retrieval models to evolving corpora over time is crucial to better align with real-world scenarios. In order to evaluate the adaptability of retrievers, we create a setup called **Dynamic Information Retrieval (DynamicIR)**. As depicted in Figure 1, our experimental setup includes three approaches: (1) *StaticIR*, where the retriever is trained on the initial corpus, (2) *Indexing-based updates*, incorporating the index of newly arrived documents into the existing index without further training on the new corpus; and (3) *Training-based updates*, where the retriever is continually pretrained on the new corpus, along with updating the index.

To conduct these experiments, we assume that we have an initial corpus $C_{initial}$ and a newly introduced corpus $C_{new}$, and datasets of query-document pairs $R_{initial}$ and $R'_{new}$ from $C_{initial}$ and $C_{new}$, respectively. Unlike $R_{initial}$, $R'_{new}$ consists of pseudo-queries, which are generated from $C_{new}$ using docT5qeury (detailed explanation is in Section 3.2). These query-document pairs are used for supervised learning. Moreover, we assess the retrieval performance with two types of evaluation sets, $Q_{initial}$ and $Q_{new}$, where the answers to the questions are within $C_{initial}$ and $C_{new}$, respectively. Some questions between $Q_{initial}$ and $Q_{new}$ are identical, except for the timestamps, necessitating the retrieval of different passages. Each

set is employed to assess the forgetting of initial knowledge and the acquisition of new knowledge.

**StaticIR.** In this part, we focus on retrieving documents only from $C_{initial}$. The training process begins with pretraining the model on $C_{initial}$ using self-supervised learning, followed by finetuning it with $R_{initial}$ using supervised learning. We evaluate it only on $Q_{initial}$ with pre-indexed $C_{initial}$.

**Indexing-based Update.** In this update setup, we incorporate the new corpus to the retrieval models by updating only the index without any parameter updates. Since we utilize a retrieval model trained in StaticIR, this updating approach is quick and straightforward. We evaluate the retriever on $Q_{initial}$ and $Q_{new}$ with pre-indexed $C_{initial}$ and $C_{new}$.

**Training-based Update.** In this advanced setup for update, we take the model pretrained on $C_{initial}$ and continually pretrain it on $C_{new}$. Subsequently, we finetune it using a combination of datasets, $R_{initial}$ and $R'_{new}$. Like indexing-based updates, we evaluate the updated retrieval model on $Q_{initial}$ and $Q_{new}$ with pre-indexed $C_{initial}$ and $C_{new}$.

In DynamicIR, we highlight the importance of striking a balance between retaining existing knowledge [21, 32] and incorporating new information. We also highly focus on computational and memory efficiency, since the practical applications like search engines handle vast and ever-changing collections of web documents, which is directly related with the practicality.

### 3.2 Benchmark

To evaluate the performance of retrieval models in a dynamic scenario, we employ StreamingQA [29] designed for temporal knowledge updates. Unlike other benchmarks on temporal retrieval [8], StreamingQA is the only benchmark that includes both the timestamps of question asked time and document publication dates, which is critical for considering the temporal dynamics. The temporal information is prepended to the text in the format of '*Today is Wednesday, May 6, 2020.* [question]' for question, and '*Thursday, February 7, 2019.* [document text]' for documents [29]. The dataset spans 14 years and includes over 50 million passages, surpassing the content size of Wikipedia used in DPR [20], which comprises 21 million passages, by over 2 times.

**Temporal Information.** StreamingQA includes a corpus spanning from 2007 to 2020, along with a supervised dataset of question-document pairs covering the years 2007 to 2019. In our work, $C_{initial}$ comprises articles from 2007 to 2019 and $C_{new}$ consists of articles from 2020. Regarding the supervised dataset, the questions in $R_{initial}$ are asked in the time range of 2007 to 2019 to query articles from this period, and the questions in $R'_{new}$ are asked in 2020 to query articles from 2020. Notably, all questions in the evaluation dataset $Q_{initial}$ and $Q_{new}$ are asked in 2020, beginning with the prefix 'Today is [Day], [Month Date] , 2020', although they query articles from 2007 to 2019 ($C_{initial}$) and 2020 ($C_{new}$), respectively.

**Pseudo-Queries for $R'_{new}$.** The original StreamingQA dataset lacks query-document pairs from $C_{new}$, making it challenging to

Soyoung Yoon, Chaeeun Kim[*], Hyunji Lee, Joel Jang, Sohee Yang, and Minjoon Seo

| Layer | Projection | Avg num of DPs |
|-------|-----------|----------------|
|       | FC1       | 1.1M           |
| FFN   | FC2       | 77K            |
|       | **Total** | **1.87M**      |
|       | Query     | 41K            |
| ATTN  | Key       | 35K            |
|       | **Total** | **76K**        |

**Table 2: Average number of Dynamic Parameters (DPs), the parameters that have large impact on acquiring new knowledge per block. DPs are significantly more prevalent in the fully connected layer, exceeding those in the attention layer.**

explore training-based updates. To address this, we generate additional 90,000 queries from $C_{new}$. To make this, we employ a trained model similar to the one used in docT5query[1] [36] for query generation. The size of this additional dataset $R'_{new}$ is similar to that of $R_{initial}$. Details of the query construction are explained in Appendix A.5.

## 4 EXPERIMENTAL SETUP

### 4.1 Retrieval Models

***Generative Retrieval (GR).*** We select SEAL [3] that employs the substrings in a passage as document identifiers and MINDER [27] that uses a combination of the titles, substrings, and pseudo-queries as identifiers. We choose the two as baselines since unlike other GR models using document IDs as identifiers [43, 44], SEAL and MINDER can be more effective on updates of individual pieces of knowledge by autoregressively generating the context using FM-index. FM-index for constrained decoding provides information on all documents in the corpus containing a specific n-gram for every decoding step, thus allowing to retrieve them [3]. Implementation details of GR are in Appendix A.2.2.

***Dual-Encoder (DE).*** We select Spider [40] and Contriever [15] as representative models for DE. Since our experiments include a pretraining phase to store the corpus itself, we use Spider and Contriever as baselines that focus on the *self-supervised methods*. Since these models do not include a supervised method, we use DPR [19] during a finetuning phase, and adhere to its original training scheme such as utilizing in-batch negative training. Thus, our DE baselines, Spider(+DPR) and Contriever(+DPR), include both pretraining and finetuning phases. Implementation details of DE are in Appendix A.2.1.

***Sparse Retrieval.*** Although our main focus are on transformer-based semantic search models to explore corpora adaptation, we also evaluate BM25 (*Lucene*)[2], a traditional retrieval model utilizing lexical matching.

[1]https://github.com/castorini/docTTTTTquery
[2]https://github.com/castorini/pyserini

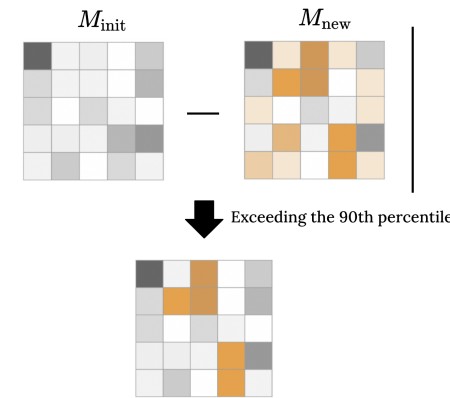

**Figure 2: Analysis on key parameters in acquiring new knowledge. Through this analysis, we identify the locations of the top 10% most activated parameters.**

### 4.2 Effective Continual Pretraining with LoRA

When continually pretraining GR on $C_{new}$, we employ LoRA [14] widely recognized for its training efficiency. To better target key parameters for incorporating new knowledge, we analyze which parameters undergo the most significant change during the acquisition of new knowledge. This analysis is inspired by works on model merging [1, 46]. We refer these crucial parameters in learning new knowledge as *Dynamic Parameters* (DPs).

To identify DPs, as illustrated in Figure 2, we follow these steps: (1) Pretrain the model on $C_{initial}$ as $M_{init}$, and continually pretrain the model on $C_{new}$ with full parameters as $M_{new}$. (2) Calculate the absolute differences in parameters between $M_{init}$ and $M_{new}$. (3) Identify parameters that exceed the 90th percentile of these absolute differences.

DPs are 2 times more prevalent in the feed-forward networks (FFN), exceeding those in the attention layer over 2 times, as shown in Table 2. This result aligns well with previous studies on the memorization of factual knowledge [6, 12]. Consequently, based on this analysis, we apply LoRA on FFN in addition to the attention layer during the continual pretraining phase. The performance is noticeably improved compared to full-parameters and conventional LoRA. In contrast to GR, DE experiences significant degradation when this approach is applied (See Appendix A.6); thus, we continually pretrain DE with full parameters. Details of the results are provided in Section 5.6.

### 4.3 Evaluation

We assess retrieval performance with three evaluation dataset, $Q_{initial}$, $Q_{new}$, and $Q_{total}$. First, we evaluate the retention of initial knowledge by 2,000 questions that should be answered from the $C_{initial}$. Second, we assess the acquisition of new knowledge by 3,000 questions that should be answered from $C_{new}$. Both sets of 5,000 questions are randomly extracted from the entire evaluation data of StreamingQA, maintaining the ratio (16.60%) of each question type for initial knowledge and new knowledge. Finally, we assess total performance by calculating the unweighted average of the

| Evaluation | Performance ($hit@5$) | | | | Efficiency | | | |
|---|---|---|---|---|---|---|---|---|
| | $Q_{total}$ | $Q_{initial}$ | $Q_{new}$ | $Q_{new}^{\text{w/o bias}}$ | Inference Flops | Indexing Time | Inference Latency ($T_{online}$ / $T_{offline}$) | Storage Footprint |
| **StaticIR** | | | | | | | | |
| Spider $_{DE}$ | - | 19.65% | - | - | 9.0e+10 | 18.9h | **24.48ms** / 26m | 173.8G |
| Contriever $_{DE}$ | - | 16.10% | - | - | 9.0e+10 | 18.9h | 212.4ms$^{\dagger}$ / 9.8m | 88.8G |
| SEAL $_{GR}$ | - | 34.95% | - | - | **4.3e+10** | **2.7h** | 545.9ms / **1m 5s** | **34.5G** |
| MINDER $_{GR}$ | - | **37.90%** | - | - | **4.3e+10** | **2.7h** | 424.6ms / **1m 5s** | **34.5G** |
| LTRGR $_{GR}$ | - | 37.85% | - | - | **4.3e+10** | **2.7h** | 424.6ms / **1m 5s** | **34.5G** |
| **Indexing-based Update** | | | | | | | | |
| Spider $_{DE}$ | 24.82% (16.5%) | 15.60% | 34.03% | 17.40% | 1.0e+11 | 20.4h | **24.84ms** / 28m | 196.8G |
| Contriever $_{DE}$ | 19.66% (11.01%) | 13.75% | 28.53% | 8.27% | 1.0e+11 | 20.4h | 228.8ms$^{\dagger}$ / 10.5m | 99.8G |
| SEAL $_{GR}$ | 33.05% (35.13%) | 32.75% | 33.50% | 37.50% | **4.3e+10** | 3.1h | 612.2ms / **1m 26s** | **37.5G** |
| MINDER $_{GR}$ | **38.63% (38.56%)** | **37.65%** | **39.70%** | **39.47%** | **4.3e+10** | 3.1h | 485.4ms / **1m 26s** | **37.5G** |
| LTRGR $_{GR}$ | 38.30% (37.47%) | 37.30% | 39.30% | 37.63% | **4.3e+10** | 3.1h | 485.4ms / **1m 26s** | **37.5G** |
| **Training-based Update** | | | | | | | | |
| Spider $_{DE}$ | 36.99% (19.58%) | 21.75% | **52.23%** | 17.40% | 1.0e+11 | 20.4h | **24.84ms** / 28m | 196.8G |
| Contriever $_{DE}$ | 23.85% (9.82%) | 8.20% | 39.50% | 11.43% | 1.0e+11 | 20.4h | 228.8m$^{\dagger}$ / 10.5m | 99.8G |
| SEAL $_{GR}$ | 41.01% (38.89%) | 38.25% | 43.77% | 39.53% | **4.3e+10** | 3.1h | 612.2ms / **1m 26s** | **37.5G** |
| MINDER $_{GR}$ | **41.54% (39.31%)** | **38.85%** | 44.23% | **39.77%** | **4.3e+10** | 3.1h | 485.4ms / **1m 26s** | **37.5G** |
| LTRGR $_{GR}$ | 41.02% (39.14%) | 38.50% | 43.53% | 39.77% | **4.3e+10** | 3.1h | 485.4ms / **1m 26s** | **37.5G** |

$^{\dagger}$ For Contriever, $T_{online}$ is measured using faiss-cpu.

Spider and Contriever are further supervised using DPR.

**Table 3: Results of DynamicIR. Our experiments are divided into 3 setups, (1) StaticIR, (2) Indexing-based updates, and (3) Training-based updates. For each setups, we assess the performance on $Q_{total}$, $Q_{total}^{\text{w/o bias}}$, $Q_{initial}$, $Q_{new}$, and $Q_{new}^{\text{w/o bias}}$ where the bias-inducing timestamps are removed. $Q_{new}^{\text{w/o bias}}$ is the average of $Q_{initial}$ and $Q_{new}^{\text{w/o bias}}$. Efficiency is evaluated using 4 metrics on the right side. For Inference Latency, $T_{online}$ indicates the time required for query embedding and search, and $T_{offline}$ represents the time for loading the indexed corpus. We highlight the best scores in bold for each setup. Additionally, the zero-shot performance for all models is provided in Appendix 7.**

above two performance. Furthermore, we measure computational and memory efficiency to comprehensively assess the practicality of retrieval models in Section 6.

## 4.4 Metric

To assess the practicality of retrieval models, we measure the retrieval performance along with the efficiency of each models. For retrieval performance, we report $Hits@5$ metric, which measures whether the gold-standard passages is included in the top 5 retrieved passages. Most document search systems do not limit results to one or provide too many; we consider 5 to be a reasonable number for assessment. Additionally, we report full results of $Hits@k$ and $AnswerRecall@k$ ($k \in \{5, 10, 50, 100\}$) in Appendix A.8. Answer Recall measures whether the retrieved passage contains an exact lexical match for the gold-standard answer. For retrieval efficiency, we report inference FLOPs (Floating Point Operations), indexing time, inference latency, and storage footprint (Details are in Section 6).

## 5 RESULTS AND ANALYSIS

In this section, we showcase the adaptability and robustness of GR and DE, and provide an analysis on utilizing $R'_{new}$ and enhanced continual pretraining approach during the training-based update. We also discuss the performance of BM25 in dynamic environments.

## 5.1 GR has greater adaptability in both update scenarios.

We define *adaptability* as the ability of retrieval models to maintain the performance after the updates, compared to the performance before the updates. To evaluate the adaptability, we examine the performance on $Q_{total}$ in each update scenario and compare it with that on $Q_{initial}$ in StaticIR (See Table 3).

First, *in indexing-based updates, GR exhibits 4% greater adaptability to new corpora compared to DE*. Specifically, when we look from $Q_{initial}$ of StaticIR to $Q_{total}$, GR maintains average performance, while DE demonstrates a 4% degradation on average. Second, *in*

Soyoung Yoon, Chaeeun Kim[*], Hyunji Lee, Joel Jang, Sohee Yang, and Minjoon Seo

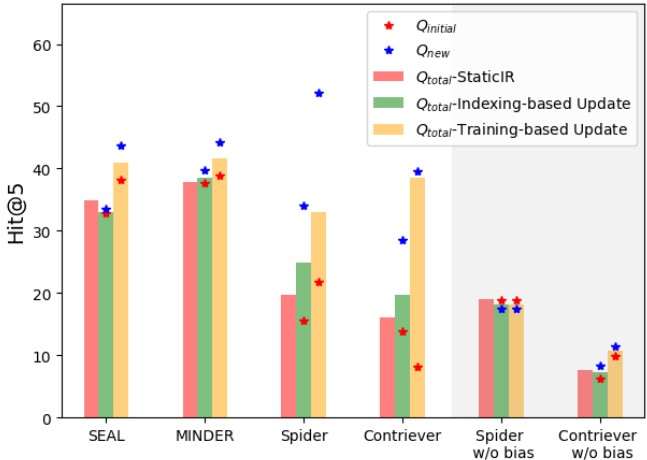

**Figure 3: Visualization of total performance in DynamicIR. The star marks highlight the change in the gap between $Q_{initial}$ and $Q_{new}$ of DE before and after the elimination of the bias-inducing factor.**

| Model | $R'_{new}$ | $Q_{total}$ | $Q_{initial}$ | $Q_{new}$ |
|---|---|---|---|---|
| Spider $_{DE}$ | with | **36.99%** | 21.75% | **52.23%** |
| | w/o | 35.77% | **29.90%** | 41.63% |
| Contriever $_{DE}$ | with | **23.85%** | 8.20% | **39.50%** |
| | w/o | 19.12% | **13.90%** | 24.33% |
| SEAL $_{GR}$ | with | **41.01%** | **38.25%** | **43.77%** |
| | w/o | 37.91% | 37.25% | 38.90% |
| MINDER $_{GR}$ | with | **41.54%** | **38.85%** | **44.23%** |
| | w/o | 37.80% | 38.15% | 40.03% |

**Table 4: Analysis the effectiveness of $R'_{new}$ with pseudo-queries in training-based updates. In this table, w/o refers only using $R_{initial}$ during finetuning. The results in hit@5 show that it is effective to include the $R'_{new}$.**

*training-based updates, GR shows 11% greater adaptability to new corpora compared to DE.* Notably, GR shows a 5% average gain in performance. On the other hand, DE demonstrates a 6% degradation on average.

For DE, we extract the update score from $Q_{total}^{\text{w/o bias}}$ instead of $Q_{total}$. Because DE exhibits a significant inherent bias towards the lexical overlap of timestamps when evaluating $Q_{new}$. We delve deeper into this phenomenon below.

### 5.2 DE shows significant bias towards temporal data.

*We observe a bias in DE towards the lexical overlap of timestamps* from the unusually high performance on $Q_{new}$ not only in training-based updates (31% higher than $Q_{initial}$) but also in *indexing-based updates* (19% higher than $Q_{initial}$) where the models never encounter new corpora during training. This phenomenon stems from the temporal information, where all timestamps in the queries and in the documents of the evaluation dataset to be retrieved are set to *the year 2020*, introducing bias towards lexical overlap. In Table 3, $Q_{new}^{\text{w/o bias}}$ shows that removing bias-inducing timestamps significantly reduces DE's performance on $Q_{new}$, bringing it to a level similar to $Q_{initial}$. See the change in the gap between $Q_{initial}$ and $Q_{new}$ before and after removing timestamps in Figure 3. Whereas, GR shows robust retrieval performance on temporal information. For more detailed explanations, refer to Appendix A.4.

### 5.3 GR better acquires new corpora even without parameter updates.

We assess the ability to acquire new knowledge through $Q_{new}$ in both update scenarios. For DE, we consider the performance of $Q_{new}^{\text{w/o bias}}$ instead of $Q_{new}$, reflecting the impact of bias as described above. In indexing-based updates, Table 3 demonstrates that *GR*

*excels in retrieving new knowledge even without parameter updates.* GR achieves a 2% higher score in $Q_{new}$ compared to $Q_{initial}$ of StaticIR. Conversely, DE shows a 2% average degradation in $Q_{new}^{\text{w/o bias}}$. Similarly, for training-based updates, while DE decreases by $2 - 5\%$ in $Q_{new}^{\text{w/o bias}}$, GR gains $6 - 9\%$ in $Q_{new}$ and $2 - 5\%$ in $Q_{new}^{\text{w/o bias}}$.

### 5.4 GR better preserves initial knowledge.

To assess the ability to retain initial knowledge, we analyze the performance on $Q_{initial}$ in both update setups, comparing it with $Q_{initial}$ in StaticIR.

For GR, Table 3 does not show notable signs of forgetting; instead, training on new corpora helps improve $Q_{initial}$ in training-based updates. We hypothesize that the GR models may be influenced by the use of language model attributes for learning language distributions. Through additional training on in-domain data, GR can gain advantages in preserving initial knowledge.

On the other hand, DE shows a $3 - 4\%$ degradation in indexing-based updates and a $0 - 8\%$ decrease in training-based updates. This observation indicates that *DE tends to forget initial knowledge more during updates compared to GR.*

### 5.5 $R'_{new}$ enhances the overall performance of GR.

We analyze the effectiveness of utilizing $R'_{new}$, query-document pairs where the queries are pseudo-queries generated from $C_{new}$ using docT5query. In addition to the results of related works [28, 31, 33, 36, 39, 44, 52], our findings on dynamic corpora demonstrate that employing $R'_{new}$ generated from new corpora is beneficial for retrieving not only new knowledge but also initial knowledge for GR (See Table 4).

We believe experiencing benefits on $Q_{initial}$ despite training with $R'_{new}$ is also attributed to the utilization of language models attributes for learning language distributions. Conversely, in the case

| Model | Continual Pretraining | $Q_{total}$ | $Q_{initial}$ | $Q_{new}$ |
|---|---|---|---|---|
| | ours (attn+ffn) | **38.08%** | **37.25%** | **38.90%** |
| SEAL $_{GR}$ | convent. LoRA (attn) | 31.69% | 32.00% | 31.37% |
| | full params | 29.92% | 28.50% | 31.33% |
| | ours (attn+ffn) | **39.04%** | **38.15%** | 39.93% |
| MINDER $_{GR}$ | convent. LoRA (attn) | 38.35% | 37.50% | 39.20% |
| | full params | 38.83% | 35.30% | **42.37%** |

**Table 5: Analysis of the effectiveness of our continual pretraining approach targeting the key parameters. The results indicate hit@5 scores for training-based updates on activating full parameters (557M params), convent. LoRA (2.4M params), and our approach (3.1M params).**

of DE, we observe a $5 - 8\%$ degradation in $Q_{initial}$, indicating forgetting. Moreover, since DE has a bias towards timestamps, if we explore $Q_{new}^{w/o\ bias}$ instead of $Q_{new}$, $R'_{new}$ would not help DE at all.

## 5.6 Applying LoRA to FFN benefits GR in both preservation and acquisition of knowledge.

Based on the analysis described in Section 4.2, our continual pretraining approach significantly improves the adaptability of GR. As shown in Table 5, activating FFN modules, which include many key parameters for adapting to new knowledge, helps not only $Q_{new}$ but also $Q_{initial}$, compared to using conventional LoRA (convent. LoRA) and full parameters (full params). Specifically, targeting the key parameters helps mitigate the forgetting issue by updating sparsely, which surprisingly is more effective than the conventional approach of updating fewer parameters in LoRA. Additionally, it maximizes the acquisition of new knowledge even more than training with full parameters, where all parameters are updated, in SEAL.

## 5.7 BM25 shows temporal bias and limited adaptability.

We investigate how BM25 performs in dynamic corpora with temporal information. Notably, BM25 surpasses transformer-based retrieval models on the StreamingQA benchmark, in contrast to its performance [3, 5, 26, 27, 45, 48] on other benchmarks such as KILT [37], MSMARCO [2] and NaturalQuestions [22] (See Table 6). However, BM25 exhibits limitations in handling temporal data, with a 4.7% degradation observed when timestamps are removed ($Q_{new} \rightarrow Q_{new}^{w/o\ bias}$ in **[Update]**), indicating a bias towards lexical matching of data from 2020. Furthermore, it struggles to maintain its initial performance, experiencing a 2.16% degradation when integrated with a new 6M corpus ($Q_{initial}$ in **[Static]** $\rightarrow$ $Q_{total}^{w/o\ bias}$ in **[Update]**). These findings underscore the need for retrieval models to move beyond textual matching, focusing not only on semantic searching [30] but also on adapting to evolving corpora and maintaining robustness across diverse data characteristics.

| Evaluation | Performance ($hit@5$) | | | | |
|---|---|---|---|---|---|
| | $Q_{total}$ | $Q_{total}^{w/o\ bias}$ | $Q_{initial}$ | $Q_{new}$ | $Q_{new}^{w/o\ bias}$ |
| **[Static]** BM25 | - | - | 43.35% | - | - |
| **[Update]** BM25 | 43.54% | 41.19% | 37.25% | 49.83% | 45.13% |

**Table 6: Performance of BM25 in StaticIR and Indexing-based Update. Through these results, we see the bias of temporal information via difference between $Q_{new}$ and $Q_{new}^{w/o\ bias}$ and adaptability through a comparison between [Static] $Q_{initial}$ and [Update] $Q_{total}^{w/o\ bias}$ which is unweighted average of $Q_{initial}$ and $Q_{new}^{w/o\ bias}$.**

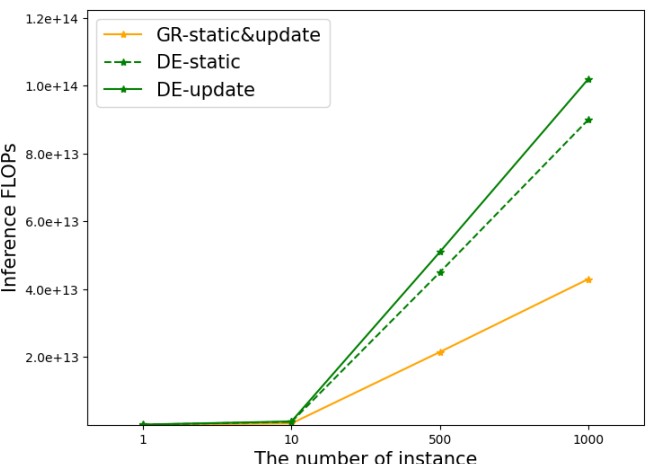

**Figure 4: Inference FLOPs according to the number of instances. The flops for GR on both the static and updated corpus are identical, as it maintains consistent flops regardless of the corpus size unlike DE.**

## 6 COMPUTATION & MEMORY EFFICIENCY

In this section, we provide the results of computational and memory efficiency. To measure indexing time and inference latency, we use an 80G A100 GPU, keeping the server empty except for our process throughout the measurement.

*Inference FLOPs.* We analyze the inference FLOPs [‡] of DE and GR to assess their computational efficiency. We approximately measure FLOPs per instance using $DE_{flops}$ for DE and $GR_{flops}$ for GR defined as below. We use the notation *IP* for inner product, *FW* for

---

[‡]FLOPs (Floating Point Operations) is the number of floating-point arithmetic calculations.

forward pass, and *Beam* for beam search.

$$DE_{flops} = FW_{flops}^{enc} + C \div Cluster \times Nearest \times IP_{flops}$$
$$GR_{flops} = FW_{flops}^{enc} + L \times Beam_{flops}$$
$$IP_{flops} = d_{\text{model}} + (d_{\text{model}} - 1)$$
$$FW_{flops} = 2N + 2n_{\text{layer}}n_{\text{ctx}}d_{\text{attn}}$$
$$Beam_{flops} = (FW_{flops}^{dec} + IP_{flops} \times |V| \log|V|) \times B$$

where $C$ is the corpus size, $L$ is the sequence length of output, $d_{model}$ is dimension of hidden vector, $N$ is the model size, $n_{layer}$ is the number of layers, $n_{ctx}$ is the length of input context, $d_{attn}$ is the dimension of attention, $V$ is the vocab size, and $B$ is the beam size. *Cluster* is the total number of centeroids (clusters), *Nearest* is the number of clusters to search, $|V| \log|V|$ is the complexity of obtaining possible token successors with FM-index [3]. We calculate $FW_{flops}$ for the transformer based on Table 1 in [18] and apply it to the encoder and decoder.

As shown in Table 3, our results reveal that *GR requires 2 times fewer computations per instance over DE*, exhibiting 4.3e+10 for the all three setups. In contrast, DE has 9.0e+10 for StaticIR and 1.0e+11 for indexing-based and training-based updates. Detailed calculations are in Appendix A.7. Figure 4 illustrates that GR offers superior efficiency as the number of instances increases. Moreover, unlike DE, which exhibits $O(N)$ complexity, where $N$ represents the corpus size, GR maintains a constant $O(1)$ complexity.

***Indexing Time.*** There is a difference in the concept of indexing between DE and GR. For DE, this involves embedding, which converts the corpus into representations using an encoder. In GR, indexing refers the data processing of document identifiers to constrain beam search decoding, ensuring the generation of valid identifiers. Note that we process data without applying sharding.

As shown in Table 3, our results exhibit that *GR (3.1h) requires 6× less time than DE (20.4h) for indexing $C_{initial}$ and $C_{new}$*. The crucial aspect of indexing is that DE necessitates re-indexing the entire corpus each time whenever the model is updated, irrespective of the corpus update. In contrast, GR has a significant advantage in that they do not require re-indexing when the model is changed. This issue becomes even more prominent when the corpus size is substantial.

***Inference Latency.*** Inference process can be divided into two stages: (1) loading a pre-indexed corpus and (2) retrieving, which includes query embedding and search. We classify the former as *offline latency* ($T_{offline}$) and the latter as *online latency* ($T_{online}$), measuring both. $T_{online}$ in Table 3 is reported for a single instance.

Table 3 shows *GR is 10 times faster than DE when retrieving from updated corpora for $T_{offline}$*. Unlike DE, which stores each passage representation in vector form, GR does not need much time to load the index since it stores knowledge within its parameters.

*For $T_{online}$, however, GR is 20 times slower than DE using faiss-gpu.* Although DE requires 2 times more inference flops, it seems that the FAISS [16] module contributes significantly to the inference speed of DE.

While online latency remains a challenge in GR, we anticipate that this can be addressed through the development of powerful computing resources or external modules like FAISS for GR in the future.

***Storage Footprint.*** We measure the storage footprint of the retrieval model and the pre-indexed corpus, which are required for performing retrieval.

Table 3 indicates that *GR has 4 less storage requirements over DE for updated corpora*. Notably, the memory requirements for DE are directly affected by the corpus size, as they store representations of all documents in vector form outside the retrieval model. In contrast, GR has minimal dependence on the corpus size by storing knowledge in its internal parameters.

GR also stores information approximately 4 times more efficiently per passage from the perspective of information theory. Specifically, we incorporate 6M $C_{new}$ to the retrieval model using only 3.1M parameters (with LoRA) and an extra 3GB FM-index in training-based updates. That is, when updating using FP16, GR requires approximately 501 bytes to store one passage, which is the sum of 1 byte and 500 bytes for the parameters and FM-index, respectively. In contrast, DE demands 2,048 bytes for storing a passage in index with a dimension of 1,024. However, we note that the index of DE is often quantized to FP8 or higher.

## 7 CONCLUSION

In this work, we conduct an extensive comparison of DE and GR, focusing on their practicality. By establishing a DynamicIR setup, we showcase how retrieval models perform in real-world scenarios where knowledge evolves over time. Although DE is more commonly utilized in practical IR systems, our findings highlight GR's superior performance in terms of adaptability, robustness, and efficiency. While online inference latency of GR remains the challenge, it has potential as a practical IR system in the future. This potential stems from GR's high adaptability to evolving knowledge, robustness in handling temporal data without introducing bias, lower memory requirements, fewer inference flops, and reduced indexing time. In this paper, we shed light on the practical advantages of GR on dynamic corpora.

## 8 LIMITATIONS

Our study has certain limitations. First, the evaluation dataset in the StreamingQA benchmark lacks diversity. All timestamps in the queries and in the documents to be retrieved from $Q_{new}$ are set to the year 2020. This matching may introduce bias towards lexical overlap of temporal information when evaluating the acquisition of new knowledge. For a more dynamic evaluation, it is better to consider diverse query timestamps. Second, due to the scarcity of datasets that reflect temporal updates, we rely only on StreamingQA. While this dataset comprises 50 million articles spanning 14 years, a more comprehensive assessment across various datasets is needed to generalize our findings. Third, our findings cover large-scale corpus updates, yet they raise the question of how these results apply across multiple update frequencies. Lastly, while our results highlight the numerous advantages of GR in terms of adaptability to new corpora, inference flops, and memory, our evaluation of online inference latency demonstrates that DE has a faster speed compared to GR, primarily due to the FAISS module.

## ACKNOWLEDGMENTS

This work was partly supported by Samsung Electronics grant (General-purpose generative retrieval, 2022, 80%) and Institute of Information & communications Technology Planning & Evaluation (IITP) grant funded by the Korea government (MSIT) (No.2022-0-00264, Comprehensive Video Understanding and Generation with Knowledge-based Deep Logic Neural Network, 20%).

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

## A APPENDIX

### A.1 Zero-shot performance of DE and GR

We conduct zero-shot experiments to assess the base performance of retrieval models on StreamingQA, utilizing Spider trained on NQ, Contriever trained on CCNet and Wikipedia, SEAL trained on KILT, and MINDER trained NQ. The results of the zero-shot experiments are presented in in Table 7.

### A.2 Implementation Details

#### A.2.1 Dual Encoder.

**Spider.** Spider experiments are conducted using 8× A100 80GB GPUs, and our implementation setup is primarily based on Spider. SPIDER [40][§] code. We employ the bert-large-uncased pretrained model (336M) from HuggingFace, with fp16 enabled and weight sharing, configuring a batch size of 512 and a maximum sequence length of 240. For the pretraining stage, we run a full epoch with a learning rate of 2e-05 and a warm-up of 2,000 steps. The pretraining data is created by running the spider code on the provided documents from StreamingQA, through which we generate synthetic queries using recurring spans. This yields 95,199,412 pretraining data from base corpus and 21,698,933 from new corpus, which are used for StaticIR and DynamicIR, respectively. It takes about 5

---

[§]https://github.com/oriram/spider

| Model | $Q_{total}$ | $Q_{initial}$ | $Q_{new}$ |
|---|---|---|---|
| SPIDER $_{DE}$ | 13.28% | 8.95% | 17.60% |
| Contriever $_{DE}$ | 18.74% | 7.15% | 30.33% |
| SEAL $_{GR}$ | 20.60% | 19.80% | 21.40% |
| MINDER $_{GR}$ | 25.87% | 25.00% | 26.73% |

**Table 7: Zero-shot performance on updated corpora. It demonstrates the zero-shot performance in hit@5 achieved without further training from released checkpoints. Overall, it exhibits a similar trend to our models trained on StreamingQA dataset.**

days for pretraining the base model and 25 hours for continual pretraining the updated model. For the finetuning stage, we run for maximum 10 epochs with learning rate of 1e-05 and warm-up of 1,000 steps with batch size of 512. We select the best checkpoint with lowest validation loss.

**Contriever.** Contriever experiments are done on 4× A100 40GB GPUs. We employ bert-large-uncased pretrained model (336M) and follow the paper [15] and their official codebase[¶] for the implementation and hyperparameter setup. We adjust the per_gpu batch size from 256 to 64 to fit in our gpu resource. Total step size is 110,000 for base (warmup 4,000 steps) and 16,000 (warmup 1,000 steps) for continual pretraining on $Cnew$, which is equivalent to one epoch. Learning rate is set to 1e-04. For the finetuning stage, we run contriever for maximum 10 epochs (about 8000 steps, warmup for 100 steps) with eval frequency of 200 steps and select the checkpoint with lowest eval loss. The per_gpu batch size is set to 32. All the hyperparemeters are the same with the pretraining setup, except the ones mentioned above.

#### A.2.2 Generative Retrieval.

**SEAL.** We employ the bart-large pre-trained model (400M) for GR and train the model in Fairseq framework for using SEAL.[3][‖]. Due to this context, when we utilize LoRA method, we implement the method within the Fairseq framework. For the pretraining stage of the base retrieval model in StaticIR, we generate 2 random spans and 1 full passage with the publication timestamp as input for each instance using the past corpus, resulting in 130,897,221 (130M) unsupervised data. We train the initial model on 16× A100 40GB GPUs with a batch size of 7,400 tokens and a learning rate of 6e-5. Subsequently, for the finetuning stage in StaticIR using $R_{initial}$, we use 10 random spans as document identifiers per question, resulting in 994,020 (994K). We train this model using 4× A100 80GB GPUs with batch size of 11,000tokens and a learning rate of 6e-5. In the continual pretraining stage for the updated model in training-based updates of DynamicIR, we use 3 random spans and 1 full passage with the publication timestamp as input for each instance, utilizing

---

[¶]https://github.com/facebookresearch/contriever
[‖]https://github.com/facebookresearch/SEAL

| MINDER $_{GR}$ | $Q_{total}$ | $Q_{initial}$ | $Q_{new}$ |
|---|---|---|---|
| w/o title | 41.54% | 38.85% | 44.23% |
| with pseudo-title | 40.86% | 38.15% | 43.57% |

**Table 8: MINDER with and without Titles as Identifiers. The results in hit@5 indicate that there is little difference between the use of identifiers with and without the title.**

| Spider $_{DE}$ | $Q_{total}$ | $Q_{initial}$ | $Q_{new}$ |
|---|---|---|---|
| Full parameters | 36.99% | 21.75% | 52.23% |
| LoRA | 26.44% | 10.05% | 42.83% |

**Table 9: Spider with and without LoRA when pretraining on $C_{new}$. The results in hit@5 show that DE achieves higher performance when pretraining with full parameters not to apply LoRA.**

the updated corpus, which results in 24,471,541 (24M) unsupervised data. We train this updated model using 4× A100 80GB GPUs with a batch size of 11,000 tokens and a learning rate of 1e-4. Subsequently for finetuning stage in training-based update of DynimicIR using $R_{initial}$ and $R'_{new}$, we generate 10 random spans as passage identifiers per question, respectively, resulting in 1,894,020(1.8M) data. During inference, we set the beam size to 10.

**LTRGR.** We use 4× A100 80GB GPUs for the learning-to-rank phase. We employ MINDER to create base models and then follow the configuration of LTRGR when learning to rank, except for setting the number of epochs and hits to 5 and 150, respectively, and omitting the title. During inference, we set the beam size to 10. When generating the training dataset for learning to rank, due to computational memory issues in processing 150 hits of our large finetuning dataset, we randomly sample 25% from $R_{initial}$ when training initial models used for StaticIR and Indexing-based update setups. For updated models, we randomly sample 25% from the combination of $R_{initial}$ and $R'_{new}$, maintaining a 1:1 ratio between them.

**MINDER.** We use 2× A100 80GB GPUs for MINDER experiments. We use the pretrained model which is used for SEAL experiments, since MINDER has identical pretraining process to that of SEAL. For retrieval model of StaticIR, we create MINDER-specific data comprising of 10 spans and 5 pseudo-queries as passage identifiers per question, resulting in 1,491,030 (1.4M). For retrieval model of training-based updates in DynamicIR, we generate 10 spans and 5 pseudo-queries, resulting in 2,841,030 (2.8M) data. We run all MINDER models for maximum 10 epochs using with max token of 18,000 and a learning rate of 6e-5. During inference, we set the beam size to 10.

### A.3 Difference in the presence of Titles as Identifiers for MINDER

The original MINDER model employs three components, titles, substrings, and pseudo-queries, as its identifiers. However, as the StreamingQA dataset lacks title information, we exclude document titles when constructing the MINDER model. To investigate the impact of this omission on performance, we conduct an analysis within training-based updates by fine-tuning utilizing pseudo-titles generated by GPT-3.5. Our results demonstrate that the omission of titles, in comparison to the utilization of pseudo-titles, has a negligible impact on performance as shown in Table 8.

### A.4 Exploration of DE's bias towards lexical overlap of timestamps

All timestamps in the queries and in the documents to be retrieved are set to the year 2020. In this context, to clarify the bias of DE towards temporal information, we finetune the models using a dataset where query dates are removed. Subsequently, we evaluate the models using an evaluation dataset where query dates are eliminated. This experiment is viable because, out of a total of 5,000 evaluation instances, only 7 cases require different documents for the same question but with different query timestamps. Through the results $Q_{new}^{\text{w/o bias}}$ in Table 10 compared to $Q_{new}$ in Table 3, we identify that the unexpectedly high performance of DE models stems from the lexical overlap with the timestamp. On the other hand, GR conducts retrievals more stably with fewer constraints on the lexical characteristics.

### A.5 Constructing the query-document pairs from new corpus

Reflecting the original evaluation dataset's distribution which balanced similar proportions of new (2020) and base (2007 – 2019) data, we replicate this distribution in our query generation based on new corpus. We randomly selected 90,000 passages from the 6 million 2020 passages. Subsequently, we finetuned a T5-base model on the query-document pairs from StreamingQA's base corpus, applying a hyperparameter configuration similar to docT5 query generation, feeding date-prefixed passages as input and producing date-prefixed queries as output. The training process comprises three epochs, with each taking roughly 45 minutes on an NVIDIA A6000 GPU. We then use the trained T5 model to generate one pseudo-query for each of the 90,000 selected passages, a process lasting approximately 90 minutes. Ensuring alignment with our study's temporal focus, we verify that the date information in the generated queries corresponded to 2020. Following a manual adjustment to ensure the queries are asked in 2020, we assemble the queries and corresponding documents into an additional finetuning dataset for the retrieval models, a process that takes about four hours in total. Examples of the finetuning dataset are in Table 11.

### A.6 Application of enhanced continual pretraining approach on DE

Unlike GR, LoRA on feed-forward network and attention layers does not improve the retrieval performance of DE. As shown in Table 9, it is evident that DE achieves higher performance when

Soyoung Yoon, Chaeeun Kim[*], Hyunji Lee, Joel Jang, Sohee Yang, and Minjoon Seo

| w/o timestamp | Indexing-based updates | | Training-based updates | |
|---|---|---|---|---|
| | $Q_{initial}^{\text{w/o bias}}$ | $Q_{new}^{\text{w/o bias}}$ | $Q_{initial}^{\text{w/o bias}}$ | $Q_{new}^{\text{w/o bias}}$ |
| Spider $_{DE}$ | 18.90% | **17.40%** | 18.90% | **17.40%** |
| Contriever $_{DE}$ | 6.25% | **8.27%** | 9.85% | **11.43%** |
| SEAL $_{GR}$ | 35.35% | 37.50% | 35.30% | 39.53% |
| MINDER $_{GR}$ | 36.85% | 39.47% | 38.45% | 43.57% |

**Table 10: Ablation Study on the bias towards temporal information. DE shows a lexical bias toward timestamps on $Q_{new}$ where all queries are asked in 2020 and the gold documents are published also in 2020. When removing the timestamp of query, the performance drastically drops, while GR does not exhibit noticeable changes.**

pretraining on $C_{new}$ with the full parameters rather than using LoRA. The degradation in hit@5 is noticeable not only in $Q_{new}$ but also in $Q_{initial}$, indicating that the application of LoRA is not beneficial for both retaining initial knowledge and acquiring new knowledge.

### A.7 Calculation Details of Inference FLOPs

We provide an approximate calculation of inference flops for DE and GR on updated corpora. For DE using the bert-large-uncased, its configurations are $N$=336M, $d_{model}$=1,024, $n_{layer}$=24, $n_{ctx}$=512, and $C$=50M. For query embedding, $FW_{flops}$ is 697M, and for searching, $C \times IP_{flops}$ is 102B. The total inference flops ($DE_{flops}$) amount to approximately 102B + 697M ≈ 102.7B. For GR using the bart-large, its configurations are $N$=400M, $d_{model}$=1,024, $n_{layer}$=12, $n_{ctx}$=1,024, V=50,265, L=10, and B=10. For the encoding process, $FW_{flops}$ is 425M,

and for the decoding process, $FW_{flops}$ is 42.5B. The total inference flops ($GR_{flops}$) amount to approximately 425M + 42.5B ≈ 43B.

Note that for DE, we employ the exhaustive (brute-force) search method adopted by our baselines. Some models can employ approximate search techniques, such as clustering, introducing a trade-off between speed and accuracy as they conduct exhaustive searches within nearby clusters.

### A.8 Full performance on Hit and Answer Recall

We present the full results of evaluating the performance of DE and GR in both StaticIR and DynamicIR (indexing-based updates and training-based updates). We employ Hit@N and Answer Recall@N metrics, where N is set to 5, 10, 50, and 100, to assess retrieval performance. The results are in Table 12 and Table 13 for Hit and Answer Recall, respectively.

Received 26 April 2024; revised 14 June 2024; accepted 2 June 2024

| Pseudo-Query | Gold Passage |
|---|---|
| Today is Sunday, October 25, 2020. When did the pay gap between Pakistani employees and white employees decrease to 2%? | Monday, October 12, 2020. In 2019 median hourly earnings for white Irish employees were 40. 5% higher than those for other white employees at 17.55, while Chinese workers earned 23.1% more at 15.38 an hour and Indian workers earned 14.43 an hour - a negative pay gap of 15.5%. Annual pay gap Breaking down the data by gender, the ONS said ethnic minority men earned 6.1% less than white men while ethnic minority women earned 2.1% more than white women. The ONS added that ethnicity pay gaps differed by age group. Ämong those aged 30 years and over, those in ethnic minority tend to earn less than those of white ethnicities,ït said. In contrast, those in the ethnic minority group aged 16 to 29 years tend to earn more than those of white ethnicities of the same age. Gender pay gap The ONS found that the pay gap of 16% for Pakistani employees aged more than 30 shrank to 2% for those aged 16-29. |
| Today is Sunday, May 2, 2020. What was the top level of the FTSE 100? | Tuesday, April 28, 2020. But the big weekly shop has made a comeback, with the amount families spend on an average shopping trip hitting a record high. The new tracking data comes after Tesco boss Dave Lewis said the pandemic had changed people's shopping habits, which he said have r̈everted to how they were 10 or 15 years ago.M̈eanwhile, is this the end of loo roll wars? Spaghetti hoops have overtaken lavatory paper as the most out-of-stock item in Britain's stores. Follow our guide to minimising your risk of catching Covid-19 while shopping. The oil giant said there would continue to be an ëxceptional level of uncertaintyïn the sector. Meanwhile, the FTSE 100 soared to a seven-week high. Follow live updates in our markets blog. |
| Today is Tuesday, March 24, 2020. Why did President Trump sign an executive order banning hoarding? | Tuesday, March 24, 2020. President Donald Trump signs executive order banning hoarding March 23 (UPI) – President Donald Trump on Monday signed an executive order to prevent hoarding and price gouging for supplies needed to combat the COVID-19 pandemic. During a briefing by the White House Coronavirus Task Force, Trump and Attorney General William Barr outlined the order which bans the hoarding of vital medical equipment and supplies including hand sanitizer, face masks and personal protection equipment. Ẅe want to prevent price gouging and critical health and medical resources are going to be protected in every form,T̈rump said. The order will allow Health and Human Services Secretary Alex Azar to designate certain essential supplies a s scarce, which will make it a crime to stockpile those items in excessive quantities. Barr said the limits prohibit stockpiling in amounts greater than r̈easonable personal or business needsör for the purpose of selling them in ëxcess of prevailing market pricesädding that the order is not aimed at consumers or businesses stockpiling supplies for their own operation. Ẅe're talking about people hoarding these goods and materials on an industrial scale for the purpose of manipulating the market and ultimately deriving windfall profits,ḧe said. |
| Today is Tuesday, November 27, 2020. What is the name of the radio channel Joe Biden was on? | Monday, November 16, 2020. 'Heal the damage': Activists urge Joe Biden to move beyond b̈order securityÄs Joe Biden prepares to take office, activists say the president-elect must not only take meaningful action to stabilize the US-Mexico border, but also reckon with his own history of militarizing the border landscape and communities. Biden has promised to end many of the Trump administration's border policies, but has yet to unveil the kind of bold immigration plan that would suggest a true departure from Obama-era priorities. Cecilia Muoz, Obama's top immigration adviser who memorably defended the administration's decision to deport hundreds of thousands of immigrants, was recently added to Biden's transition team. Biden has stated that he will cease construction of the border wall, telling National Public Radio in August that there will be n̈ot another foot of wall,änd that his administration will close lawsuits aimed at confiscating land to make way for construction. His immigration plan will also rescind Trump's declaration of a n̈ational emergencyön the southern border, which the Trump administration has used to siphon funds from the Department of Defense to finance construction, circumventing Congress in an action recently declared illegal by an appeals court. Some lawmakers along the border find these developments heartening, after Trump's border wall construction has devastated sensitive ecosystems, tribal spaces, and communities, and has been continuously challenged in court. |

**Table 11: Examples of Finetuning dataset $R'_{new}$ created by docT5.**

Soyoung Yoon, Chaeeun Kim[*], Hyunji Lee, Joel Jang, Sohee Yang, and Minjoon Seo

| Model | Method | hit@5 | | | hit@10 | | | hit@50 | | | hit@100 | | |
|-------|--------|-------|---------|------|--------|---------|------|--------|---------|------|---------|---------|------|
| | | Total | initial | New | Total | initial | New | Total | initial | New | Total | initial | New |
| Spider | StaticIR | 19.65 | 19.65 | – | 25.40 | 25.40 | – | 38.20 | 38.20 | – | 44.50 | 44.50 | – |
| | Index–based Update | 24.82 | 15.60 | 34.03 | 30.67 | 20.20 | 41.13 | 44.92 | 32.80 | 57.03 | 51.28 | 38.45 | 64.10 |
| | Train–based Update | 36.99 | 21.75 | 52.23 | 43.74 | 26.95 | 60.53 | 58.75 | 40.40 | 77.10 | 64.84 | 46.95 | 82.73 |
| Contriever | StaticIR | 16.10 | 16.10 | – | 20.25 | 20.25 | – | 33.80 | 33.80 | – | 40.90 | 40.90 | – |
| | Index–based Update | 21.14 | 13.75 | 28.53 | 25.17 | 17.35 | 36.90 | 39.44 | 29.45 | 54.43 | 46.26 | 35.65 | 62.17 |
| | Train–based Update | 23.85 | 8.20 | 39.50 | 29.26 | 10.55 | 47.97 | 43.66 | 20.35 | 66.97 | 49.64 | 25.35 | 73.93 |
| SEAL | StaticIR | 34.95 | 34.95 | – | 41.80 | 41.80 | – | 57.25 | 57.25 | – | 63.10 | 63.10 | – |
| | Index–based Update | 33.13 | 32.75 | 33.50 | 39.64 | 38.90 | 40.37 | 54.14 | 54.50 | 53.77 | 59.71 | 60.55 | 58.87 |
| | Train–based Update | 41.01 | 38.25 | 43.77 | 47.99 | 45.30 | 50.67 | 62.90 | 60.20 | 65.60 | 67.79 | 65.00 | 70.57 |
| MINDER | StaticIR | 37.90 | 37.90 | – | 45.00 | 45.00 | – | 59.60 | 59.60 | – | 64.00 | 64.00 | – |
| | Index–based Update | 38.68 | 37.65 | 39.70 | 45.27 | 44.40 | 46.13 | 60.87 | 60.60 | 61.13 | 66.13 | 66.35 | 65.90 |
| | Train–based Update | 41.54 | 38.85 | 44.23 | 48.29 | 45.60 | 50.97 | 63.12 | 60.80 | 65.43 | 68.43 | 66.25 | 70.60 |

**Table 12: Full results on the Hit of DE and GR.**

| Model | Method | answer recall @5 | | | answer recall @10 | | | answer recall @50 | | | answer recall @100 | | |
|-------|--------|------------------|---------|------|-------------------|---------|------|-------------------|---------|------|--------------------|---------|------|
| | | Total | initial | New | Total | initial | New | Total | initial | New | Total | initial | New |
| Spider | StaticIR | 37.55 | 37.55 | – | 47.45 | 47.45 | – | 67.65 | 67.65 | – | 74.80 | 74.80 | – |
| | Index–based Update | 44.24 | 33.45 | 55.03 | 52.93 | 41.50 | 64.37 | 70.77 | 61.70 | 79.83 | 76.68 | 69.20 | 84.17 |
| | Train–based Update | 55.79 | 41.05 | 70.53 | 64.32 | 49.90 | 78.73 | 79.25 | 68.90 | 89.60 | 83.63 | 75.50 | 91.77 |
| Contriever | StaticIR | 28.90 | 28.90 | – | 37.60 | 37.60 | – | 60.20 | 60.20 | – | 68.25 | 68.25 | – |
| | Index–based Update | 31.34 | 25.15 | 40.63 | 41.84 | 34.80 | 52.40 | 63.05 | 55.15 | 74.90 | 70.98 | 64.30 | 81.00 |
| | Train–based Update | 37.14 | 20.15 | 54.13 | 46.54 | 28.15 | 64.93 | 66.21 | 48.65 | 83.77 | 72.33 | 56.85 | 87.80 |
| SEAL | StaticIR | 58.25 | 58.25 | – | 66.30 | 66.30 | – | 80.45 | 80.45 | – | 83.60 | 83.60 | – |
| | Index–based Update | 55.85 | 56.80 | 54.90 | 63.68 | 64.45 | 62.90 | 77.58 | 78.95 | 76.20 | 81.49 | 82.75 | 80.23 |
| | Train–based Update | 62.44 | 59.95 | 64.93 | 70.25 | 68.10 | 72.40 | 81.65 | 80.30 | 83.00 | 85.02 | 84.10 | 85.93 |
| MINDER | StaticIR | 59.50 | 59.50 | – | 68.10 | 68.10 | – | 80.35 | 80.35 | – | 83.75 | 83.75 | – |
| | Index–based Update | 54.23 | 54.45 | 54.00 | 62.96 | 63.75 | 62.17 | 76.54 | 78.00 | 75.07 | 79.79 | 81.20 | 78.37 |
| | Train–based Update | 56.74 | 55.35 | 58.13 | 64.45 | 63.70 | 65.20 | 77.19 | 77.40 | 76.97 | 80.34 | 80.50 | 80.17 |

**Table 13: Full results on the Answer Recall of DE and GR.**