# OpenReview forum: "Exploring the Practicality of Generative Retrieval on Dynamic Corpora"
_ACM.org/SIGIR/2024/Workshop/Gen-IR — Gen-IR_SIGIR24_

### Official Review · Reviewer_UccD · 2024-05-24
**Despite some minor concerns, this paper can provide remarkable insights and is above the acceptance threshold.**

**Rating:** 1
**Confidence:** 4

**Review:**

This paper focuses on the continual learning scenario of dual encoders and generative retrievals, extensive empirical results demonstrate that generative retrieal methods are more adaptable, robust and efficient.

Strengths:
* This paper focuses on the dynamic corpora scenario of generative IR methods, which is a significant and challenging problem faced in real-world retrieval scenarios.
* The experimental setup is  rigorous and reasonable, and the evaluation metrics are extensive. Moreover, the experimental analysis is section 5 is highly detailed and clear, making this paper even solid.
* The paper is well-structured and well-written, and the proposed findings in secion 1 and limitations in section 8 can provide clear and informative insights to researchers.

Weaknesses:
* There are some related works very close to this paper that has been omitted and should be mentioned.
  - Continual Learning for Generative Retrieval over Dynamic Corpora.
  - Continually Updating Generative Retrieval on Dynamic Corpora.
  - CorpusBrain++: A Continual Generative Pre-Training Framework for Knowledge-Intensive Language Tasks
* As demonstrated in Table 1, only one incremental phase (2020) is considered, the setup might be more solid with more incremental phases in consideration.
* In real-world search engines, whether the new query-document pairs ($R_{new}^{'}$) are available in the incremental phase is in doubt, and I'm more inclined to think that we can't get labeled quey-document pairs in the incremental stage.

---

### Official Review · Reviewer_L7Xo · 2024-05-25
**Review of Exploring the Practicality of Generative Retrieval on Dynamic Corpora**

**Rating:** -1
**Confidence:** 4

**Review:**

## Summary
This paper presents a comprehensive eval of Generative Retrieval (GR) compared to Dual Encoders (DE) in dynamic information retrieval scenarios. They establish DynamicIR that utilizes the StreamingQA benchmark to examine the adaptability, robustness, and efficiency of the four models (2 GR, 2 DE). The findings demonstrate that GR showcases better effectiveness when faced with evolving knowledge, handling temporal data more robustly, and achieving lower inference costs and memory efficiency compared to DE.

## Strengths
1) Accounts for various facets like inference FLOPS, indexing time, storage, etc. during the evaluation of models, often missed in literature.
2) By using the StreamingQA benchmark, the paper addresses the practical challenges of IR systems dealing with constantly evolving corpora, making the findings highly relevant for real-world search applications.
3) Well-situated comparison of the two paradigms.
4) The results and analysis section is fairly comprehensive.
## Weaknesses
1) The paper reads very much like a draft which affects the quality and clarity. For instance "[2] David R. Cheriton. 2019. From doc2query to docTTTTTquery." in citations is incorrect and additionally, there is a table in between References.
2) DynamicIR feels very close to the explorations from Mehta et al. 23, which this paper cites very sparsely, even though it explores a lot of similar questions and with more detail (like aspects of unlearning documents etc.) Use of QGEN for changing environments is also explored there.
3) Exploring only a single dataset (StreamingQA) that is Wikipedia-based limits the generalizability of the findings across different IR contexts.
4) Aren't SPIDER and Contriever in their formulation insufficient to lead to some of the conclusions? They should ideally be pretrained with synthetic queries and the sort that are obvious modifications and also employed by the Generative Retrieval methods? What about simply using BM25, this should be included to better situate results too.
5) Single Hit@5 retrieval effectiveness metric can be limiting.

---

### Official Review · Reviewer_4V9X · 2024-05-25
**This paper evaluates generative retrieval models in dynamic information retrieval scenarios**

**Rating:** 2
**Confidence:** 4

**Review:**

This paper investigates both the retrieval models with dual encoders architecture as well as the encoder-decoder-based generative retrieval models, in the context of dynamic IR. In particular, this work evaluates the adaptability, robustness, and efficiency in terms of both eh computational and memory usage.

Strengths:
1. This work is meaningful. This paper conducted an extensive evaluation of DE and DR models under a practical scenario where the corpus evolves over time. How to update the retrieval model with the newest knowledge is a realistic consideration.
2. This paper also provides a detailed analysis of various aspects of the retrieval models, including retrieval performance, adaptability, robustness, and efficiency. The generated insights are useful to the community.

Weakness:
1. there are only two models selected for each family, i.e DE and Generative retrieval models. However, for each retrieval paradigm, there are many models proposed. For instance, in the DE paradigm, the authors only experimented on the single-vector models, I would be curious if the observations can be extended to the multiple-vector search models.
2. From Table 2, it seems there is no difference between the "training-based update" vs. "indexing-based update", maybe the authors can make some point on this observation.

---

### Decision · Program_Chairs · 2024-05-28

**Decision:**

Accept

**Comment:**

This paper evaluates generative retrieval models in the context of an evolving corpus. Reviewers found this to be an interesting problem and appreciated the extensive experiments. However, they also noted closely related work that has been minimized or omitted from the paper, which should be addressed in the camera ready.